# Atypically Shaped Cardiomyocytes (ACMs): The Identification, Characterization and New Insights into a Subpopulation of Cardiomyocytes

**DOI:** 10.3390/biom12070896

**Published:** 2022-06-27

**Authors:** Mariko Omatsu-Kanbe, Ryo Fukunaga, Xinya Mi, Hiroshi Matsuura

**Affiliations:** Department of Physiology, Shiga University of Medical Science, Otsu 520-2192, Shiga, Japan; ryofuku@belle.shiga-med.ac.jp (R.F.); mixinya@phar.kyushu-u.ac.jp (X.M.); matuurah@belle.shiga-med.ac.jp (H.M.)

**Keywords:** atypically shaped cardiomyocytes, ACMs, subpopulation of cardiomyocytes, spontaneous beating, fetal cardiac gene proteins, cell fusion, ischemic resistance, cardiomyocyte, cardiac ventricle, heart

## Abstract

In the adult mammalian heart, no data have yet shown the existence of cardiomyocyte-differentiable stem cells that can be used to practically repair the injured myocardium. Atypically shaped cardiomyocytes (ACMs) are found in cultures of the cardiomyocyte-removed fraction obtained from cardiac ventricles from neonatal to aged mice. ACMs are thought to be a subpopulation of cardiomyocytes or immature cardiomyocytes, most closely resembling cardiomyocytes due to their spontaneous beating, well-organized sarcomere and the expression of cardiac-specific proteins, including some fetal cardiac gene proteins. In this review, we focus on the characteristics of ACMs compared with ventricular myocytes and discuss whether these cells can be substitutes for damaged cardiomyocytes. ACMs reside in the interstitial spaces among ventricular myocytes and survive under severely hypoxic conditions fatal to ventricular myocytes. ACMs have not been observed to divide or proliferate, similar to cardiomyocytes, but they maintain their ability to fuse with each other. Thus, it is worthwhile to understand the role of ACMs and especially how these cells perform cell fusion or function independently in vivo. It may aid in the development of new approaches to cell therapy to protect the injured heart or the clarification of the pathogenesis underlying arrhythmia in the injured heart.

## 1. Introduction

The mammalian heart is one of the organs with a low regeneration capacity after birth [1,2]. In the development stage, embryonic cardiomyocytes arise from the early cardiac progenitors or the proliferation of pre-existing cardiomyocytes [3]. Neonatal cardiomyocytes undergo additional rounds of DNA synthesis without cytokinesis, resulting in the binucleation or multinucleation shortly after birth [4], and have transient regeneration potential during this period [5]. In the adult heart, a number of studies have reported evidence to support the notion that the pre-existing cardiomyocytes are capable of re-entering the cell cycle in both the human and mouse heart [6,7,8,9,10], the rate and degree of cardiomyocyte renewal under both physiological and pathophysiological conditions are far too small; thus, dead cells are not substantially replaced by renewed cells [11,12]. In the mouse heart, the turnover of cells through the proliferation of resident cardiomyocytes is estimated to occur at a rate of approximately 1.3–4% per year, whereas in the human heart, the annual turnover rate of cardiomyocytes is observed to gradually decrease from 1% at 25 years of age to 0.45% at 75 years of age; thus, the rate of cardiomyocyte exchange in adulthood is <1% per year [6,13].

Following myocardial infarction, the heart loses approximately 25% of the cardiomyocytes within a few hours [14]; however, it is estimated that <0.1% of cardiomyocytes re-enter the cell cycle [15]. At present, the “proliferative activity” of the postnatal mammalian heart is thought to be limited to the process of multinucleation and polyploidization, which occurs in the early postnatal period and in the failing heart [12]. Under pathophysiological conditions, active cardiomyocyte proliferation to cardiac hypertrophy has been reported based on an autopsy of patients with LEOPARS syndrome [16]. One of the most effective tools for understanding the mechanism of binucleation, polyploidization and cell-cycle arrest of cardiomyocytes is now thought to be a model system using induced pluripotent stem cells (iPSCs) that reproduces the essential factors in the heart [17]. In addition, the upregulation of resident or bone marrow-derived progenitor cells has been reported to give rise to cardiomyocytes for repairing the heart [18,19,20,21,22,23,24,25].

As the regulation of cardiomyocyte regeneration has been one of the most important themes in clinical research, a number of studies have attempted to manipulate endogenous progenitor cells to induce differentiation into functional cardiomyocytes. The first adult resident cardiac stem cells were identified by the expression of stem cell receptor kinase (c-kit) but not blood lineage markers (Lin), c-kit^+^/Lin^−^ cells, in 2003 [26]. Thereafter, the identification of cardiac stem or progenitor cells based on characteristics such as the expression of stem cell antigen-1 (Sca-1) [27,28] and LIM-homeobox transcription factor (islet 1) [29] and the ability to exclude Hoechst 33342 dye [30], has been reported. During this period, these cells were demonstrated to differentiate into cardiomyocytes in vitro in response to hormones or chemicals, such as 5′-azacytidine [27,28], oxytocin [28,30] and trichostatin A [30]. Adult cardiac stem cells isolated by c-kit^+^/Lin^−^ selection have been shown to more systematically give rise to cardiomyocytes cultured in leukemia inhibitory factor (LIF)-deprived basic differentiation medium supplemented with several chemicals in sequence, including oxytocin and activin A [31]. Negative views have also been reported, with one review reporting that no reliable data have shown the existence of cardiomyocyte-differentiable stem cells in the adult heart that might be used in practical therapeutics to repair the injured myocardium [32]. Although there are still some limitations that hinder cell therapy using cardiac stem or progenitor cells, the methodology is constantly being innovated, with methods such as proteomic analyses of cardiac progenitor cells, including cells isolated based on surface-marker selection, differentiated human embryonic stem cells (ESCs) and iPSCs [33], and the 3D structure applications composed of cardiac and endothelial progenitor cells [34]. These methods have been accelerated with the aim of translation into cell therapy [1,2].

We detect spontaneously beating cells with peculiar morphologies expressing cardiac-specific proteins but not stem cell markers in the culture of interstitial non-myocyte fraction cells obtained from adult mouse cardiac ventricles and named them “atypically shaped cardiomyocytes (ACMs)” [35]. Similar to cardiomyocytes, ACMs do not actually proliferate. These cells have also been observed mixed in with isolated ventricular myocytes in the studies of other groups [36,37]. Native ACMs are found in the interstitial spaces among ventricular myocytes in both mice and humans [38] and are considered to be a subpopulation of cardiomyocytes or immature cardiomyocytes rather than cardiac stem or progenitor cells. However, although ACMs are able to survive in the myocardium from the neonatal period to the aged period while retaining their ability to develop into a beating cell [39], the fate of ACMs both in vitro and in vivo still remains unclear.

In this review, we focus on the differences in the morphology, protein expression and cellular function between in ACMs and cardiomyocytes to explore the identity of ACMs and consider the physiological function in vivo and their potential utility in treatment.

## 2. Variety of Heart Cells

The heart is a complex organ comprising multiple cell types, each of which plays an important role under both physiological and pathophysiological conditions [40]. Cardiomyocytes play a major role in maintaining blood circulation via their spontaneous activity and pumping function, but they do not account for the majority of cardiac cells existing in the entire heart. Indeed, cardiomyocytes have been estimated to constitute only 30–40% of total cardiac cells [41,42,43,44,45,46], indicating that a large number of cells, called “non-myocytes”, exist in the interstitial spaces. The non-myocyte group comprises heterogeneous cell lineages, such as endothelial cells, vascular smooth muscle cells, pericytes, fibroblasts, macrophages and other types of cells, such as cardiac stem or progenitor cells. These cells communicate with each other not only by direct physical contact but also by paracrine signaling [40].

### 2.1. Cardiac Endothelial Cells

Endothelium lines the interior surface of blood vessels and lymphatic vessels, demonstrating heterogeneity in its structure and function [47]. In the heart, endothelial cells can be divided into two types—those localized in the cardiac endothelium and those in the coronary vascular endothelium—that contribute to not only structural roles in cardiovascular homeostasis and angiogenesis but also the regulation of post-infarction remodeling via interaction with cardiomyocytes. Cross-interaction of endothelial cells and cardiomyocytes via paracrine signaling is necessary for cardiac development and regeneration [40,46,48,49,50]. Representatively, vascular endothelial growth factor (VEGF) is a key factor secreted by cardiomyocytes that can modulate the growth of blood vessels. The deletion of VEGF leads to not only a reduction in the number of coronary microvessels but also thinning of the ventricular wall, contractile dysfunction and an abnormal response to adrenergic stimulation [51].

### 2.2. Mural Cells

In the vascular system, the major compartment includes endothelial cells, smooth muscle cells and pericytes [40]. Vascular smooth muscle cells maintain the structure of the vessel wall, and their plastic nature enables the regeneration of the injured or diseased heart. Pericytes are vascular mural cells residing in the microvascular basement membrane that enwrap and support the microvessels and play an essential role in vascular remodeling [52,53,54,55]. The origin of the mural cells in the heart has been shown to be epicardial cells [56,57], but a recent study demonstrated that endocardial endothelial cells also function as a reservoir providing progenitors for mural cells [58].

### 2.3. Cardiac Macrophages

Tissue-resident macrophages are phagocytic immune cells that contribute to proper cardiac development and also control local homeostasis in response to diverse changes in microenvironments [59,60]. The heart contains several kinds of macrophages, which can be identified by the expression of specific markers [61]. Cardiac macrophages that originated from embryonic cells are replenished by in situ proliferation, but those derived from bone marrow cells are replenished by monocyte seeding from the circulation, playing diverse roles in the clearance of damaged cells and remodeling after myocardial infarction [61,62,63,64,65].

### 2.4. Cardiac Fibroblasts

Fibroblasts, classified as a component of connective tissue, synthesize and decompose extracellular matrix collagens and play a critical role in wound healing. Fibroblasts had been believed to constitute between 27 and 50% of the total cells in mouse and rat cardiac ventricles, respectively [41,43]. However, a recent study shows that the reevaluation of the endothelial cell population may reveal that the proportion of fibroblasts among the total cells of the heart is approximately 10%, which is lower than previously thought [45,66]. Under physiological conditions, cardiac fibroblasts are sparsely distributed in the interstitial spaces among cardiomyocytes to maintain the structure of the myocardium and the correct function of the heart via the production and degradation of the extracellular matrix [67]. Under pathophysiological conditions, however, cardiac fibroblasts rapidly proliferate and become activated, and thereafter a portion of these activated fibroblasts further differentiate into myofibroblasts. Myofibroblasts abundantly express α-SMA and other proteins to play a key role in the development of scar tissue after cardiac injury, and in the injury resolution stage, myoblasts gradually become more quiescent, reverting to a resting, senescent or apoptotic fibroblast state [66,68,69,70,71,72].

## 3. Cardiac Stem or Progenitor Cells

Historically, the types of cell populations have been classified into “static”, “transit” and “stem” cell groups [73]. Adult tissue cells are thus generally thought to comprise quiescent stem cells that can proliferate and differentiate into progenitors that possess the ability to differentiate into terminal cell types, and recently, the cellular function of stem cells has been demonstrated to decline with aging [74]. In the heart, the existence and capacity of the regeneration of quiescent “stem” and/or “transit” progenitor cells have been long controversial.

Endogenous adult cardiac stem or progenitor cells were originally identified as interstitial c-kit^+^/Lin^−^ cells in 2003 [26]. Since then, a number of studies, both in vitro and in vivo, have been undertaken to verify and clarify cardiac stem cells showing self-renewal, clonogenicity and multipotency. Some target antigens, such as Sca-1 [27,28] and Islt-1 [29], have attracted attention in efforts to identify “available” cardiac progenitor cells that might be used for cardiac regeneration, as in studies focusing on c-kit. Most such efforts are based on the immuno-phenotype definition, which is used to sieve and concentrate specifically marked cells. Another approach is to isolate cardiac side population cells identified by their ability to exclude DNA-binding Hoechst33342 dye [30]; this unique ability was found in the side population cells in a variety of organs, including bone marrow and heart [75]. These progenitor cells were demonstrated to differentiate into cardiomyocytes in vitro in response to hormones or chemicals, such as 5′-azacytidine [27,28], oxytocin [28,30] and trichostatin A [30], and also home to the injured heart after in vivo transplantation [27,30]. Consequently, c-kit^+^/Lin^−^ cells resident in the heart have become one of the most promising targets for cell therapy [31,76,77,78,79]. The regenerative activities of these cardiac stem cells have been demonstrated under ischemic injury [80,81], pressure overload [81], and overdose of isoproterenol [82]. In contrast, some studies have reported that cardiac stem cells minimally contribute to cardiomyocytes in the adult heart or have more assertively reported that the adult heart lacks an endogenous functional pool of myogenic precursor cells [83,84,85,86]. Further techniques using cardiac stem or progenitor cells, such as proteomic and Glyco(proteo)mic analyses [33] and 3D culture in combination with the niche stem cells [34], are underway with the aim of developing practical cell therapies.

Cardiac fibroblasts are generated from epicardial and endocardial epithelial cells through the epithelial-to-mesenchymal transition and the endothelial-to-mesenchymal transition, respectively [87,88]. Single-cell transcriptional profiling studies reveal that mouse cardiac fibroblasts comprise heterogeneous lineages [89,90]. Colony-forming unit-fibroblast (CFU-F) assay, a method for determining one of the characteristic features of stem cells by examining the colony-forming ability of mesenchymal cells in the culture environment, revealed that CFU-Fs exist in the heart [91]. The cardiac CFU-Fs display similar properties to those of bone marrow mesenchymal stem cells, but lineage tracing studies have demonstrated that cardiac and bone marrow-derived CFU-Fs have different lineage origins. It is reported that cardiac fibroblasts can be directly reprogrammed into functional cardiomyocytes in mice using the developmental transcription factors Gata4, Mef2c and Tbx5 [92], thereby suggesting possible approaches for regeneration after cardiac injury. Subsequently, the direct reprogramming of cardiac fibroblasts in vivo has been reported by the injection of a cocktail of transcription factors, such as GMT and the combination of GMT and Hand2 [93,94]. However, cardiac fibroblasts in the injured heart are thought to originate from diverse cell lineages, and the population of cardiac fibroblasts undergoing successful direct reprogramming into cardiomyocytes in vivo still remains small [95]. Further improvement is considered necessary before the direct cardiac reprogramming method can be applied to human therapeutics [96].

## 4. Atypically Shaped Cardiomyocytes (ACMs)

We found beating cells in the three-dimensional (3D) culture of cardiomyocyte-removed fraction cells using a methylcellulose-based semi-solid medium. The isolation of cardiomyocytes is performed via coronary perfusion of the whole heart with enzymes using either the retrograde [97] or antegrade [36,98] perfusion method, and the cardiomyocyte-depleted fraction including interstitial heart cells of various cell lineages can then be cultured in semi-solid culture medium. After tightly adhering to the bottom of the appropriate dish, the cells grow over several days and start spontaneously beating. Given its peculiar morphology (Figure 1), we named these cells ACMs [35]. ACMs are also observed in the cell culture with a commonly used liquid culture medium [35], but the number of ACMs, especially beating cells, is extremely low. These beating cells have occasionally been observed as distorted or unusual cells present in culture dishes of isolated cardiomyocytes, where they have not received particular attention [36,37]. Observations indicate that 3D culture is a better condition for the development of these cells in comparison to two-dimensional (2D) culture with a liquid culture medium, which is considered one reason why these cells have not received much attention thus far.

ACMs can be detected in cultures prepared from neonatal to aged mice without showing appreciable proliferation during long-term culturing [39]. The data thus indicate that ACMs survive in the heart for a life-long period but lack a self-renewing or clonogenic ability. A few days after plating, ACMs already show the automaticity and express characteristic proteins for ventricular and atrial myocytes, SA nodal pacemaker cells and fetal cardiac cells, indicating that the cells undergo the differentiation process to become cardiomyocytes as opposed to having multipotency. Therefore—at least based on the present data—it is unlikely that these would be classified as progenitor cells; rather, they would be classified as a subpopulation of cardiomyocytes or immature cardiomyocytes.

## 5. Characteristics of ACMs

### 5.1. Peculiar Morphology

One of the most obvious characteristics of ACMs, in addition to their spontaneous contracting activity, is that the cell shapes are markedly different from those of normal rod-shaped cardiac ventricular myocytes (Figure 1). ACMs possess a spherical shape at the start of culture, but after a few days of culture, they show peculiar shapes with many branches and/or protrusions, and most possess multiple nuclei with bulge(s) on the cell surface (Figure 1). However, while the cell shapes are complicated, the sarcomere structures are well organized and regularly arranged up to the branched tips [35,36,37]. The protrusions of ACMs not only come out horizontally, touching the bottom of the culture dish, but also grow three-dimensionally; thus, these cells are thicker than the proliferating fibroblast-like cells observed in the same culture dish, indicating that the cytoplasm is enriched in the contractile proteins. As the tight adherence of the plasma membrane to the culture dish may be a limitation to forming cells, the cell shapes in the heart tissues are not clear.

### 5.2. Automaticity

In most beating ACMs, the rhythmic action potentials are recorded that are reversibly suppressed by acetylcholine [35], while isoproterenol also shortens the peak intervals of the spontaneous Ca^2+^ transient in ACMs [99]. These findings are similar to the cell responses detected in sino-atrial (SA) node pacemaker cells [100,101], showing that the receptors and signal induction molecules for autonomic nervous systems function properly in ACMs. Unlike SA node pacemaker cells, ACMs often exhibit abnormal electrical automaticity caused by the repeated events of marked hyperpolarization to some extent and the subsequent depolarization. These arrhythmic events occur naturally, and the frequency varies from cell to cell.

It would be interesting to learn whether or not ACMs residing in the heart show automaticity under both physiological and pathophysiological conditions. The heart possesses a highly specialized system for generating rhythmic impulses to cause rhythmic contraction of the cardiomyocytes and for conducting these impulses rapidly throughout the heart. The electrical coupling within cardiomyocytes in ventricles is facilitated by gap junctions mainly composed of connexin 43 (Cx43) localized at the intercalated discs, whereas SA nodal cells do not express this type of gap junction proteins [102,103]. The expression of Cx43 in ACMs is low and particularly detected in the peri-nuclear area at the beginning of the culture before gradually increasing and spreading towards the cell periphery from five to eight days of culture along with the morphological maturation, confirmed by the expression of contractile protein [104]. Similar observations have been reported in which changes in the distribution of Cx43 in cardiomyocytes are observed in both cultured cells [105] and also in the myocardium of individuals with compensated and decompensated cardiac hypertrophy [106]. In addition, ACMs show an average maximal diastolic potential of around −65 mV [35], while the resting membrane potential in the isolated ventricular myocytes obtained from adult mice using our preparation method is approximately −74 mV on average [107], suggesting that the ACMs are not likely to electrically stimulate neighboring cardiomyocytes. These data suggest that the small native ACMs resident in the interstitial spaces have no—or very small—effects on the ventricular myocytes, at least in a healthy heart.

### 5.3. Protein Expression

While ACMs are small cells, with a diameter of approximately 10 µm immediately after isolation from the adult cardiac ventricular tissues, the large amounts of contractile proteins, such as α-actinin (ACTN) (Figure 1) and cardiac troponin T (cTnT), are synthesized within several days to help them increase their length more than 10-fold. Interestingly, ACMs, isolated from cardiac ventricular tissues, typically express hyperpolarization-activated cyclic nucleotide-gated channel 4 (HCN4), T-type voltage-gated Ca^2+^ channel (Ca_V_3.2), and atrial natriuretic peptide (ANP) (Table 1). It should be noted that these proteins are not functionally expressed in normal adult ventricles and are referred to as fetal cardiac gene proteins.

Adult cardiac stem cells, which were originally found in the c-kit^+^/Lin^−^ cardiac cells subpopulation, possessing the characteristics of self-renewal, clonogenicity and multipotency, can regenerate functional myocardium in vivo [26]. Eliminating cells expressing endothelial and hematopoietic markers, such as CD31 and CD45, has been imperative for the isolation of cardiac stem cells from the heart [31,79]. ACMs do not show appreciable proliferation during long-term culturing and simultaneously express characteristic proteins of ventricular and atrial myocytes, SA nodal pacemaker cells and fetal cardiac cells, suggesting that these cells have different properties from cardiac stem cells or stem cells of other types. As expected, typical stem cell surface markers, Sca-1, c-kit, hematopoietic stem cell marker (CD34), platelet-endothelial adhesion molecule-1 (CD31) and vascular endothelial cell growth factor receptor 2 (Flk-1), are absent in ACMs (Table 1). The evidence suggests that ACMs should not be classified as stem or progenitor cells that can undergo the differentiation process.

### 5.4. Ischemic Tolerance

Since ACMs are cultured in a methylcellulose-based semi-solid medium, it is more difficult to perform medium exchange than with a liquid medium, so a small droplet of the liquid medium is added from time to time during long-term culture. However, the proliferation of fibroblasts renders the culture condition an acidic one of chronic malnutrition. Surprisingly, ACMs in a medium that has turned a yellowish-orange color continue contracting rhythmically despite the harsh culture environment for over one month. Furthermore, another important feature of ACMs during long-term culture is that they fuse with each other while maintaining spontaneous contraction.

Cardiomyocytes are highly sensitive to hypoxia; for example, in the human heart, ventricular myocytes become irreversibly damaged approximately 30 min after blood flow stops [68,71]. Following the reestablishment of the blood flow, the reperfusion of the coronary artery paradoxically causes further—occasionally lethal—damage to the heart and as well as a wide range of organs [108]. ACMs can survive these lethal ischemic conditions by blocking airflow by covering the cell suspension with oil [109], with half of the cells subsequently able to develop into beating cells, while approximately 85% of ventricular myocytes die within 90 min [99]. ACMs have thus been shown to possess ischemic resistance in addition to resistance to malnutrition and acidic conditions, suggesting that these cells may survive after cardiomyocytes die in injured hearts. The finding that ACMs (Prp^+^/cTnT^+^ cells, as stated in Section 6) survived in the peripheral area of infarction in pathological heart tissue specimens obtained from the patients who had a myocardial infarction [104] supports this view.

### 5.5. Constitutively Active Autophagy

Autophagy is an evolutionally conserved process for the degradation of long-lived and/or damaged proteins and organelles occurring in cells throughout the body [110,111,112] and during the neonatal period in particular, thus providing a necessary source of energy in various tissues [113]. Under physiological conditions, the autophagy activity in the heart remains low, aiding in the maintenance of the cell components, and is rapidly activated under pathophysiological conditions to support cardio-protection [114,115,116,117,118,119]. However, during the only early neonatal starvation period, autophagy in the heart is known to be activated [113].

In contrast, autophagy is constitutively activated in ACMs, not only to support cellular functions, but it also plays an essential role in the development of beating cells in the culture [99]. Constitutively active autophagy is thought to enable ACMs to rapidly synthesize proteins to grow into large beating cells and to continue beating in harsh environments by providing a source of energy.

### 5.6. Multinucleation and Cell Fusion

The most obvious characteristic of ACMs is that approximately 76% of these cells have multiple nuclei (Figure 1), sometimes more than four [38]. In mice, the majority of cardiomyocytes undergo additional DNA synthesis without cytokinesis within approximately 14 days after birth, resulting in multinucleation [3,12]. However, we have not observed such nuclear fission in ACMs during culturing; thus, one possible explanation for the multiple nuclei of ACMs is that cells fuse with each other, whether it occurs in vivo or after culturing.

“Native ACMs” that are resident in the heart tissues (as stated in Section 6) have often been observed as clusters [38], potentially resulting in the formation of fused cells with plural nuclei through the cell preparation procedure. Among the various interstitial cells that exist in the culture dish, ACMs are only able to fuse with the same type of cell [38]. When other types of cells are attached, ACMs pull the cell membrane along while beating, but they do not fuse with any cells other than ACMs [99]. These phenomena indicate that ACMs can only fuse with cells that have a cell membrane with the same membrane components, leading to the hypothesis that cardiomyocytes are indeed candidates for fusion with ACMs. Unfortunately, the coculture of ACMs and ventricular myocytes has not succeeded because isolated ventricular myocytes cannot survive in culture while ACMs settle at the bottom of the culture dish and grow. Some ACMs are observed to closely contact with ventricular myocytes but not attach, and then ventricular myocytes shrink and die over time. Therefore, the behavior of native ACMs should be explored in the future.

## 6. Methods for Identifying Native ACMs

Specifying the origin of ACMs is also an important theme to be resolved. Although not completely elucidated, the facts are that ACMs possess all of the characteristics of ventricular and atrial myocytes and SA node pacemaker cells and survive in neonatal to older hearts with the preserved expression of fetal cardiac gene proteins [35,39] suggest that ACMs originated from immature fetal heart cells. The yield and condition of ACMs obtained from the heart are always much better with the successful isolation of ventricular myocytes, indicating that the separation of native ACMs from the neighboring cells triggers the development of these cells into beating cells. The data may suggest that native ACMs in vivo can survive and grow when these cells become independent from the microenvironment due to the death of adjacent cells in patients who suffer myocardial infarction [104]. Based on the view that ACMs are likely to exist close to ventricular myocytes in the interstitial space in the myocardium, we sought to find native ACMs in the heart.

### 6.1. Combination of Cellular Prion Protein and Contractile Protein as Markers for ACMs

ACMs can be easily identified in the culture several days after plating based on their peculiar shape and automatic beating activity, but there is no way to identify these cells using specific markers in order to isolate them from a mixture of interstitial cells. In the early developing heart, cardiac transcription factors, such as NK2 transcription factor related, locus 5 (Nkx2.5) and T-box transcription factor (Tbx) 5, are first expressed in the anterior lateral plate mesoderm, and genes encoding cardiac-specific structural proteins are subsequently expressed in the cardiac crescent [120]. However, little is known about proteins that serve as the cell surface markers in the postnatal heart, as cardiomyocytes can be easily identified by their morphological features, such as rod-like shape and sarcomere structure. Identifying specific markers of ACMs, classified as cardiomyocytes, in the myocardium is thus expected to be extremely difficult. However, this problem must be overcome in order to clarify the nature of these cells in vivo.

Cellular prion protein (Prp), also known as CD230, is ubiquitously expressed in mice, except in some specific tissues, such as the liver [121], and can be used to mark nascent cardiomyocytes and cardiac progenitors and cardiogenic populations in differentiating embryonic stem cells (ESCs) [92]. Given that Prp is strongly expressed on the cell membrane of ACMs from just after isolation before development into beating cells, those cells co-expressing Prp and contractile protein cardiac troponin T (cTnT) existing in the interstitial spaces among ventricular myocytes are considered native ACMs [38,104]. Prp^+^/cTnT^+^ cells have been shown to exist sparsely in the interstitial space of myocardial tissues in both mouse [38] and human [104] healthy heart tissue sections. While this approach is a time-consuming way of detecting double-immune-positive cells and an indirect method of identifying ACMs, there was no better way than this, at that time, to identify ACMs, especially in human pathological specimens.

### 6.2. Fetal Cardiac Specific Gene Proteins for Marking ACMs

ACMs express fetal gene proteins, such as ANP [122,123] and CaV3.2 (Table 1) [124,125], which are potential cell marker candidates. Both ANP and Cav3.2 are preferentially expressed in the fetal heart but not in adult cardiac ventricles and show an increased protein expression in the diseased heart. ANP may have utility as a marker for ACMs since it is absent in the ventricular myocytes in the postnatal healthy heart. However, particular care must be taken, as there is evidence that ANP is activated in heart failure [126,127] and also expressed in macrophages [128]. We are currently conducting analyses involving ACMs marked with these fetal cardiac-specific genes.

## 7. Future Outlook

Since the discovery of ACMs in the culture, the question arose as to what cell lineage ACMs should be classified into. One possibility was that ACMs are damaged ventricular myocytes. Cardiomyocytes that are damaged during isolation contract irregularly due to the Ca^2+^ overload and eventually die; ACMs develop into complicated and larger beating cells, which indicates that ACMs are not cardiomyocytes with damage. Incomplete or unnecessary cells are usually considered to be removed via the apoptotic pathway in the developing heart [129]. However, ACMs are not thought to be cells that are destined to die, as they can survive for the entire life of the heart or for a long-term period in culture while resisting harsh environmental conditions. The next possibility was that ACMs function as cardiac progenitors, such as satellite cells in skeletal muscle. In skeletal muscles, quiescent myoblasts, known as satellite cells, function as a source of skeletal muscle progenitor cells in the regenerative process of injured muscles, which is a well-confirmed cellular function [130]. However, we did not observe any proliferation of these cells after over a month of culture, leading us to conclude that ACMs actually lack the proliferative ability. ACMs were then determined to be cardiomyocytes that have already undergone differentiation not only due to the absence of stem cell marker proteins but also their spontaneous development into beating cells with the expression of cardiac-specific gene products. The data thus suggest that there may be some as-yet-unknown reason for the existence of these cells.

For several years since the discovery of stem cell marker-positive cells in the heart, a number of studies have reported the development of new types of cell therapy utilizing these cells [11,23,27,30,131]. The purpose of these studies was to regenerate myocardium not only via the differentiation of stem or progenitor cells into new cardiomyocytes but also by the fusion of these cells with damaged cardiomyocytes; their application in medical treatment has not yet been achieved. ACMs are observed to selectively fuse with cells of the same type during culture [38,99], which strongly suggests that these cells can fuse to damaged ventricular myocytes in the injured heart. This fusion ability may be a key clue for clarifying the role of the ACMs in the heart.

A number of therapeutic methods, including cellular therapies using regenerated cells, such as iPSCs [132,133], and acellular therapies using cell-free factors, such as exosomes, microRNAs and antibodies, have been actively studied to overcome heart diseases, and both approaches have their own advantages and disadvantages [134]. It is highly likely that undifferentiated fetal heart cells are the ancestors of ACMs, with the ability to undergo cell fusion and survive for a life-long period. If native ACMs fuse with damaged cardiomyocytes in vivo, it may restore their function. However, conversely, they may also, unfortunately, be the cause of the arrhythmias that often occur after myocardial infarction.

Overall, it would be worthwhile to find a way to consider ACMs as a subpopulation of cardiomyocytes for a new type of cell therapy or to clarify the pathogenesis of ACMs-induced arrhythmias. We hope that this review will help researchers in cardiovascular and other fields to learn about the existence of ACMs and promote interest in starting studies on their cell function and utility.

## Figures and Tables

**Figure 1 biomolecules-12-00896-f001:**
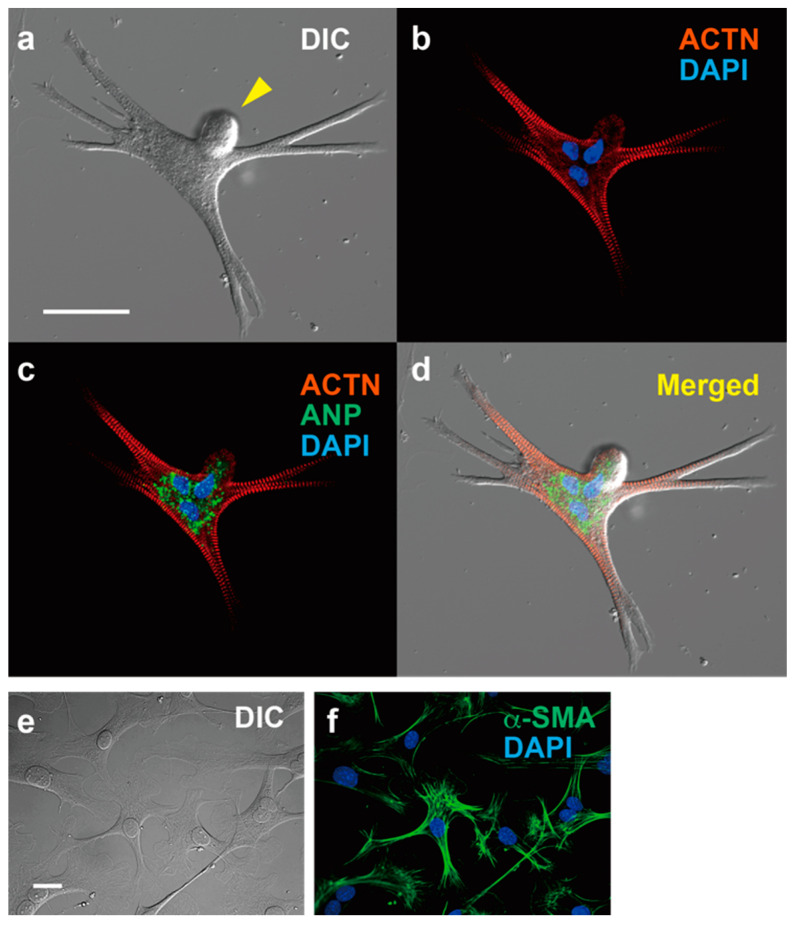
Immunofluorescent microscopy of ACM and cardiac fibroblasts. expressing ACTN and ANP, and cardiac fibroblasts expressing α-SMA. (**a**–**d**) ACM, bar, 50 µm. (**a**) Differential interference contrast image (DIC). (**b**) Immunostaining for α-actinin (ACTN, red) and DAPI staining for nuclei (blue). (**c**) Merged image for ACTN, DAPI and atrial natriuretic peptide (ANP, green). (**d**) Merged images for (**a**–**c**). (**e**,**f**) Cardiac fibroblasts, bar, 50 µm. (**e**) DIC. (**f**) Immunostaining for α-smooth muscle actin (α-SMA) and DAPI staining.

**Table 1 biomolecules-12-00896-t001:** Protein expression in ACM.

Cardiac Proteins	Protein Expression	Stem Cell Markers	Protein Expression
ACTN	Positive [35]	Sca-1	None [35]
cTnT	Positive [38]	c-kit	None [35]
Cx43	Positive [39]	CD45	None [35]
HCN4	Positive [35]	CD34	None [35]
Ca_V_3.2	Positive [39]	CD31	None [35]
ANP	Positive [39]	Flk-1	None [35]

Protein expression was determined by immunostaining for the desired protein in ACMs. ACM—atypically shaped cardiomyocyte; ACTN—α-actinin; cTnT—cardiac troponin-T; Cx43—connexin 43; HCN4—hyperpolarization-activated cyclic nucleotide-gated channel 4; Ca_V_3.2—T-type Ca^2+^ channel; ANP—atrial natriuretic peptide; Sca-1—stem cell antigen-1; c-kit—stem cell factor receptor; CD45—leukocyte common antigen; CD34—muscle stem marker; CD31—platelet-endothelial adhesion molecule; Flk-1—vascular endothelial cell growth factor receptor 2.

## Data Availability

Not applicable.

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
