# Peer review of "Atypically Shaped Cardiomyocytes (ACMs): The Identification, Characterization and New Insights into a Subpopulation of Cardiomyocytes"

_biomolecules, 2022, doi:10.3390/biom12070896_

Round 1

Reviewer 1 Report

I have no comments

Author Response

We would like to thank the Reviewer 1 for the very kind help in improving our manuscript.

Reviewer 2 Report

The authors answer crear to all my concerns

Author Response

We would like to thank the Reviewer 2 for the very kind help in improving our manuscript.

This manuscript is a resubmission of an earlier submission. The following is a list of the peer review reports and author responses from that submission.

Round 1

Reviewer 1 Report

The paper entitled “Atypically-shaped Cardiomyocytes (ACMs): The identification, characterization and new insights into cardiac progenitor cell” describes a newly identified cardiac progenitor cell population that has the potential to regenerate cardiac tissue. The theory on the presence of ACMs and their characterization, is not validated by many groups. And all the characterizations have been made by a few groups. On the other word, it is their view, not all the scientific communities’ view. That is why it might not be an appropriate topic for a scientific review.

There are also some other concerns which originates from the abovementioned reason:

  1. In the abstract, they stated that these cells are not complete cardiomyocytes. What do they mean by "they are not complete cardiomyocytes". This phrase is not scientific. They can use "they have some non-cardiomyocyte characteristics" or "they do not show some characteristics of cardiomyocytes"
  2. There is a belief that cardiac progenitor cells are present in the heart. I did not find any comparison of ACMs with these cells. My take is that they are the same. Having atypical-shape does not make them a special different type of progenitor cells. Being progenitor gives all these properties to a cell.
  3. In section 3.2., they stated that “in addition, ACMs show an average maximal diastolic potential of around -65 mV [24], while the resting membrane potential in the isolated ventricular myocytes obtained from adult mice using our preparation method is approximately -74 mV on average [81], suggesting that the ACMs are not likely to be electrically stimulated by neighboring cardiomyocytes.” This is not correct. If ACMs’ resting membrane potential is more positive, it is more likely to be stimulated by neighboring cells.
  4. In section 3.2., CD31 is not a marker of muscle stem cell.
  5. References are old. For instance, reference 1-6 which comprise the main body of their theory, are old.

Author Response

[Responses to Reviewer 1]

We would like to thank Reviewer 1 for the comments and useful criticisms on our manuscript. We have revised the manuscript in accordance with the suggestions of Reviewer 1. We feel that the suggested revisions are very appropriate and that, thanks to these suggestions, the manuscript has now been considerably improved.

We have attempted to address the points raised by Reviewer 1 in the following way:

Major comments:

We agree with the comments of Reviewer 1 that the study of ACMs has not been validated by many groups and that the characterization has been made by a few groups. To allow other groups to easily prepare ACMs, we have described the methods for the preparation and culturing of these cells in detail in previous papers. We assume that one of the reasons why it has not been realized is that the physiological importance of these cell has not been clarified. Furthermore, most research groups are not interested in progenitor cells that lack proliferative activity because they may not be used for regeneration.  

We think that undifferentiated fetal heart cells are the ancestors of ACMs with the ability to undergo cell fusion and survive for a life-long period. The data suggest two possible roles of native ACMs. One is a beneficial role that restores the function of the injured heart, for example fusing with the damaged cardiomyocytes, which may lead a new approach for cell therapy. The other is a cause of arrhythmias in the damaged heart, which often occur after myocardial infarction, and which may lead to the exploration of a new pathogenesis of arrhythmia in the injured heart. At present, we do not know which of these roles they play.

We are proceeding with our research in a step-by-step manner in order to clarify the physiological role of these cells. Recently, in collaborating with other new groups, we have succeeded in marking these cells with an atrial myocyte-specific intracellular protein tagged with GFP and are exploring the behavior of these cells in both healthy and diseased cardiac ventricles (unpublished data). 

Accordingly, the purpose of this review article is to send a message to wide audience, with the aim of increasing the number of groups that might consider ACM as a new research subject. 

Other concerns:

1) The phrase “they are not complete cardiomyocytes” in Abstract

We agree with the comments of Referee 1 that the phrase “they are not complete cardiomyocytes” is not scientific. We think that this cell itself is not a mature ventricular or atrial nor SA nodal cell, as each ACM simultaneously expresses some characteristics of ventricular myocytes, atrial myocytes and SA nodal cells. However, this explanation is not needed in the Abstract and we have therefore deleted this phrase.

2) Comparison of ACMs with cardiac progenitor cells

We agree with Reviewer 1 that the comparison of ACMs with the cardiac progenitor cells should be described in the text.

We have therefore inserted the following paragraph in Section 2.4, entitled “Atypically-shaped cardiomyocytes (ACMs)”.

“The first identified cardiac stem or progenitor cells, c-kit+/Lin- cells, have been characterized by self-renewal, clonogenicity and multipotency, and their ability to regenerate functional myocardium in vivo (Beltrami et al., 2003). ACMs do not show appreciable proliferation during long-term culturing, indicating a lack of self-renewing or clonogenic ability. A few days after plating, ACMs already show the automaticity and express characteristic proteins for ventricular and atrial myocytes, SA nodal pacemaker cells and fetal cardiac cells, indicating that the cells overcome the differentiation process to be cardiomyocytes rather than have multipotency. Therefore, —at least based on the present data—it is unlikely that these would be classified as stem cells. Rather, they would be classified as a kind of progenitor or immature cardiomyocytes.”

3) Automaticity

We would like to thank Reviewer 1 for pointing out the mistakes.

We have therefore revised the two sentences in paragraph 2 of Section 3.2, entitled “Automaticity”, as follows.

“ACMs are not likely to electrically stimulate neighboring cardiomyocytes. These data suggest that the small native ACMs resident in the interstitial spaces have no—or very small—effects on the ventricular myocytes, at least in the healthy heart.”

4) Marker proteins

We would like to thank Reviewer 1 for pointing out our mistakes. We have corrected the names of the antigens and revised paragraph 2 of Section 3.3 entitled, “Protein expression”, as follows.

“Adult cardiac stem cells, which were originally characterized by c-kit+/Lin-, possessing the characteristics of self-renewal, clonogenicity and multipotency, can regenerate functional myocardium in vivo (Beltramiet al., 2003). Eliminating cells expressing endothelial and hematopoietic markers, such as CD31 and CD45 has been imperative for the isolation of cardiac stem cells from the heart (Vicinnanza et al., 2017; Cianflone et al., 2020). ACMs do not show appreciable proliferation during long-term culturing and simultaneously express characteristic proteins of ventricular and atrial myocytes, SA nodal pacemaker cells and fetal cardiac cells, suggesting that these cells have different properties from cardiac stem cells or stem cells of other types. As expected, typical stem cell surface markers, Sca-1, c-kit, hematopoietic stem cell marker (CD34), platelet-endothelial adhesion molecule-1 (CD31) and vascular endothelial cell growth factor receptor 2 (Flk-1), are absent in ACMs (Table 1). The evidence suggests that ACMs should not be classified as stem cells that can overcome the differentiation process but rather as a kind of cardiac progenitor cell or immature cardiomyocyte.” 

The following references have been added.

Vicinanza et al. Cell Death Differ 2017, 24, 2101-2116.

Cianflone et al. Int J Mol Sci 2020, 21.

5) References and the first part of the Introduction.

I agree with Reviewer 1 that the references in the original manuscript, especially the main body of our theory, are old.

We have therefore revised paragraph 1 of Section 1, entitled “Introduction”, as follows.

“The mammalian heart is one of the organs that cannot actually regenerate after birth due to its minimal ability of self-proliferation (Alonaizan and Carr, 2022; Mehanna et al., 2022). In the development stage, embryonic cardiomyocytes arise from the early cardiac progenitors or the proliferation of pre-existing cardiomyocytes (Zhao et al., 2020). Neonatal cardiomyocytes undergo additional rounds of DNA synthesis without cytokinesis, resulting in the binucleation or multinucleation shortly after birth (Bishop et al., 2021) and have transient regeneration potential during this period (Porrello et nal., 2011). In the adult heart, a number of studies have reported evidence to support the notion that the pre-existing cardiomyocytes are capable of re-entering the cell-cycle in both the human and mouse heart (Bergman et al., 2009; Mollova et al., 2013; Senyo et al., 2013; Ali et al., 2014; Kimura et al., 2015), the rate and degree of cardiomyocyte renewal under both physiological and pathophysiological conditions are far too small; thus, dead cells are not substantially replaced by renewed cells (Laflamme and Murry, 2011; Ponnusamy et al., 2017). In the mouse heart, the turnover of cells through the proliferation of resident cardiomyocytes is estimated to occur at a rate of approximately 1.3–4% per year, whereas in the human heart, the annual turnover rate of cardiomyocytes is observed to gradually decrease from 1% at 25 years of age to 0.45% at 75 years of age; thus, the rate of cardiomyocyte exchange in adulthood is <1% per year (Bergman et a., 2009; Bergman et al., 2015).”

The following references have been added to this paragraph.

Alonaizan and Carr, Biochem Soc Trans 2022, 50, 269-281.

Bishop et al. J Am Heart Assoc 2021, 10, e017839.

Mehannna et al. World J Stem Cells 2022, 14, 1-40

Bergman et al. Cell 2015, 161, 1566-75.

Porrello et al. Science 2011, 1078-1080.

Zhao et al. Front Cell Dev Biol 2020, 8 Pages 594226

We have also cited the recent references in the text.

We would like to sincerely thank Reviewer 1, once again, for the very kind help in improving our manuscript.

Reviewer 2 Report

Dear authors, In light of the Anwersa c kit controversy, I am not convinced of the existence of ACM, as presented. There is not enough feedback from the scientific community during the 13 years since publication. Despite hope for finding the progenitor and regeneration holy grail, I cannot support this review, rather I suggest searching for better markers and descriptions, which would be cited and recognized.

Described cell type, referred to being uncovered already in 2009, has 9 citations since 2009, out of these 5 are auto citations, and only two independent authors cite the cells in other works. Pubmed giving relevant results on https://pubmed.ncbi.nlm.nih.gov/?term=Atypically-shaped+Cardiomyocytes&sort=date&size=200

Also stating, "there is no way to identify these cells using specific markers in order to isolate them from a mixture of interstitial cells" is strange and do not give any supporting material for  later discussion on Prp+/cTnT+

In light of Anwersa c kit controversy, I am not convinced of the existence of ACM, as the round is the majority of cells after dissociation, with absenting complete CMs structure, is disputable.

Despite I could agree that there has to be some progenitor cell type, I am not convinced that ACM is the right way to go, which is reflected by the lack of citing publications, with an inappropriately high number of auto citations. I cannot recommend increasing the autocitation rate and suggest rejection of the presented paper, where the revision is not an option, as the original ACM topic itself is a major flaw.

Reviewer 3 Report

The review manuscript by Omatsu-Kanbe nicely described our current understanding of the atypical shaped cardiomyocytes. The authors nicely introduce the cellular composition of the developing and adult heart and subsequently they focused on the current identification, characterization and plausible therapeutic applicability of these cells.

Overall the manuscript is interesting, yet the current characterization of these cells have only been addressed by their group, limiting thus its impact on the scientific community.

Minor points

Figure 1. It would be nice to provide similar staining for a cardiac interstitial fibroblasts, to see their cell shape differences

The subheading 3.4. Gene expression, provides limited or no value at all. Please consider re-editing and/or deleting. No detailed information of gene expression is provided and GO analyses is very superficial, similarly providing no added value.

The authors stated: “Fibroblasts are believed to account for approximately 10% of the total cells in the adult mouse heart [51].” However, reference 51 provides no information about fibroblast abundance in the adult heart, and secondly, to my point of view, the abundance of cardiac fibroblast is much greater that 10%. Please modify accordingly.

See e.g. 1: Pinto AR, Ilinykh A, Ivey MJ, Kuwabara JT, D'Antoni ML, Debuque R, Chandran A, Wang L, Arora K, Rosenthal NA, Tallquist MD. Revisiting Cardiac Cellular

Composition. Circ Res. 2016 Feb 5;118(3):400-9. doi: 10.1161/CIRCRESAHA.115.307778. Epub 2015 Dec 3. PMID: 26635390; PMCID:

PMC4744092.

Minor grammatical issues

These sentences make no sense (the red part has no link to the previous part of the sentence). Please modify accordingly

Lanes 38-40 At present, the "proliferative activity" of the postnatal mammalian heart is thought to be limited to the process of multinucleation and polyploidization, which occurs in the early postnatal period and in the failing heart [8], although an isolation technique for sorting live cardiomyocytes undergoing cytokinesis derived from human induced pluripotent stem cells (iPSCs) was reported recently [11].

Lanes 114-117 Under physiological conditions, resident mature cardiac fibroblasts are sparsely distributed in the interstitial spaces among cardiomyocytes to maintain the structure of the myocardium and the correct function of the heart via the production and degradation of extracellular matrix, which shows a low proliferation ability without the expression of  alpha-smooth muscle actin (alpha-SMA) [33,55].

Lanes 124-125 Please modify since it makes no sense. Cardiac fibroblasts can reportedly be directly reprogrammed into functional cardiomyocytes in mice, using the developmental transcription factors Gata4, Mef2c and Tbx5…..

Reviewer 4 Report

The focus of the review by Omatsu-Kanbethe and colleagues is the characterization of Atypically-shaped Cardiomyocytes (ACMs) compared with ventricular myocytes. ACMs are described as a type of cardiac progenitors identified in the cultures of cardiomyocyte-removed fraction obtained from mouse cardiac ventricles that spontaneously develop into beating cells within 3–5 days under certain condition in vitro. Moreover, the authors claim that ACMs are able to cell fusion in vivo and this discovery for them could be used as tool to develop new therapeutic approaches.

However, I have some observations that needed to be addressed before the review publication.

Major points:

  • Line 50. The authors claim that stem cells are able to differentiate into cardiomyocytes without mentioning adequate data about the potential of endogenous stem/progenitor cells to differentiate into cardiomyocytes. They only mentioned the elusive role of these cell in cardiac regeneration. Please, provide correct references in order to better describe a yet controversial field in the regenerative medicine.
  • Line 130-143. This paragraph is completely wrong. The authors show a lack of knowledge of the topic. Endogenous cardiac stem/progenitor cells have been identified for the first time in 2003 by the presence of receptor tyrosine kinase c-kit (doi: 10.1016/s0092-8674(03)00687-1). Since 2003 many research groups have documented that cardiac stem cells CSCs are clonogenic, self-renewing and multipotent, giving rise to a minimum of three different cardiogenic cell lineages (myocytes, smooth muscle and endothelial cells) both in vitro and in vivo (please see : Jia-Qiang He et al. 2011; doi.org/10.1093/cvr/cvq292; doi.org/10.3389/fcvm.2017.00047; doi.org/10.1038/cdd.2017.130; doi.org/10.1007/978-3-030-24108-7_8) and exhibit significant cardiac tissue regenerative capacity under different condition (please see: doi.org/10.1038/nm1618; doi.org/10.1073/pnas.0502678102; doi.org/10.1038/s41419-019-1655-5; doi.org/10.3390/ijms21217927). In contrast to these data, a few studies using a wrong genetic fate mapping approach ( please see : doi.org/10.1038/nature25771; doi.org/10.1161/CIRCRESAHA.114.304676; doi: 10.1016/j.phrs.2017.06.012) to purportedly fate-map c-kit-expressing or Sca-1-expressing cardiac stem cells in vivo, have claimed that cardiac stem cells minimally contribute CMs in the adult heart or more assertively that the adult heart lacks an endogenous functional pool of myogenic precursor cell, which could effectively replenish cardiomyocytes in the adult life (doi: 10.1161/CIRCULATIONAHA.119.045566.; doi.org/10.1038/nature13309; doi.org/10.1038/ncomms9701). On this premise, the Introduction section should be accordingly completely revised together with the relative reference list.
  • Line 223-224. Several reports have largely demonstrated that in order to isolate from the heart a cardiac population of cells with stem cell property it is imperative to eliminate cell-expressing endothelial and hematopoietic markers such as CD31 and CD45. The CD31pos CD45pos cell population do not have stem cell properties (doi.org/10.1038/cdd.2017.130; doi.org/10.1038/nature25771; doi.org/10.3390/ijms21217927).
  • The authors describe ACMs as immature cardiomyocytes. Typically an immature cardiomyocyte is  Why their cardiomyocytes are plural nucleated? Please clarify.

Minor points:

  • Legend of table 1 must be correctly formatted.

Round 2

Reviewer 1 Report

1. Please replace "overcome" with "undergo" in the following sentence:

A few days after plating, ACMs already show the automaticity
and express characteristic proteins for ventricular and atrial myocytes, SA nodal pacemaker cells and fetal cardiac cells, indicating that the cells overcome the differentiation process to be cardiomyocytes rather than
have multipotency.

2. please "that can overcome the differentiation process" from the following sentence:

The evidence suggests that ACMs should not be classified as stem cells that can overcome the differentiation process but rather as a kind of cardiac
progenitor cell or immature cardiomyocyte.

Reviewer 4 Report

In the introduction section the authors reported: "The mammalian heart is one of the organs that cannot actually regenerate after birth...".  Otherwise they talk about a certain turnover of cells in the heart after birth. Thus, I would say that the word "cannot" it is inadequate. Please, change the sentence as follow for example: "The mammalian heart is one of the organs with a low regenerate capacity after birth."

Thanks